# DIVA: DOMAIN INVARIANT VARIATIONAL AUTOENCODER

**Maximilian Ilse**[†]**, Jakub M. Tomczak**[†]**, Christos Louizos**[†,‡] **& Max Welling**[†,c]
[†]Amsterdam Machine Learning Lab, University of Amsterdam
[‡]TNO, Intelligent Imaging
[c]CIFAR
{m.ilse,j.m.tomczak,c.louizos,m.welling}@uva.nl

## ABSTRACT

We consider the problem of domain generalization, namely, how to learn representations given data from a set of domains that generalize to data from a previously unseen domain. We propose the domain invariant VAE (DIVA), a generative model that tackles this problem by learning three independent latent subspaces, one for the class, one for the domain and one for the object itself. In addition, we highlight that due to the generative nature of our model we can also incorporate unlabeled data from known or previously unseen domains. This property is highly desirable in fields like medical imaging where labeled data is scarce. We experimentally evaluate our model on the rotated MNIST benchmark where we show that (i) the learned subspaces are indeed complementary to each other, (ii) we improve upon recent works on this task and (iii) incorporating unlabelled data can boost the performance even further.

## 1 INTRODUCTION

Deep neural networks (DNNs) led to major breakthroughs in a variety of areas like computer vision and natural language processing. Despite their big successes recent research shows that DNNs learn the bias present in the training data. As a result they are not invariant to cues that are irrelevant to the actual task (Azulay & Weiss, 2018). This leads to a dramatic performance decrease when tested on data from a different distribution with a different bias (Torralba & Efros, 2011).

In domain generalization the goal is to learn representations from a set of similar distributions, here called domains, that can be transferred to a previously unseen domain during test time. A common motivating application, where domain generalization is crucial, is medical imaging (Blanchard et al., 2011; Muandet et al., 2013). For instance, in digital histopathology a typical task is the classification of benign and malignant tissue. However, the preparation of a histopathology image includes the staining and scanning of tissue which can greatly vary between hospitals. Moreover, a sample from a patient could be preserved in different conditions. As a result, each patient data could be treated as a separate domain (Lafarge et al., 2017; Ciompi et al., 2017). Another problem commonly encountered in medical imaging is class label scarcity. Annotating medical images is an extremely time consuming task that requires expert knowledge. However, obtaining domain labels is surprisingly cheap, since hospitals generally store information about the patient (e.g., age and sex) and the medical equipment (e.g., manufacturer and settings). Therefore, we are interested in extending the domain generalization framework to be able to deal with additional unlabeled data. We hypothesize that additional unlabeled data can lead to better domain generalization.

In this paper, we propose to tackle domain generalization via a new deep generative model that we refer to as the domain invariant variational autoencoder (DIVA). We extend the variational autoencoder (VAE) framework (Kingma & Welling, 2013) by introducing independent latent representations for an object (e.g., an image), a class label and a domain label. Such partitioning of the latent space will encourage and guide the model to disentangle these sources of variation.

Finally, by virtue of having a generative model we can also naturally handle the semi-supervised scenario, similarly to Kingma et al. (2014). We evaluate our model on a version of the MNIST dataset where each domain corresponds to a specific rotation angle of the digits.

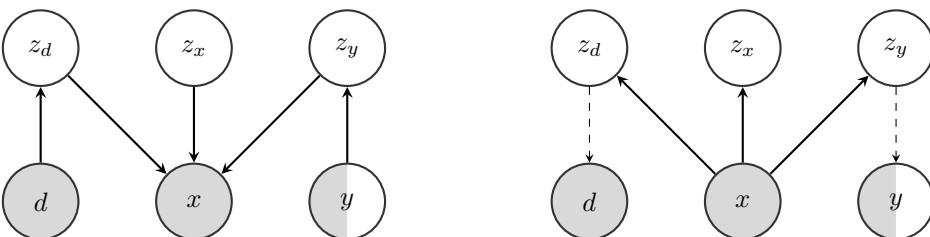

Figure 1: Left: Generative model. According to the graphical model we obtain $p(d, x, y, z_d, z_x, z_y) = p_\theta(x|z_d, z_x, z_y)p_{\theta_d}(z_d|d)p(z_x)p_{\theta_y}(z_y|y)p(d)p(y)$. Right: Inference model. We propose to factorize the variational posterior: $q_{\phi_d}(z_d|x)q_{\phi_x}(z_x|x)q_{\phi_y}(z_y|x)$. Dashed arrows represent auxiliary classifiers.

## 2 Towards domain generalization with generative models

### 2.1 Domain generalization

We follow the domain generalization definitions used in (Muandet et al., 2013). A domain is defined as a joint distribution $p(x, y)$ on $\mathcal{X} \times \mathcal{Y}$, where $\mathcal{X}$ denotes an input space and $\mathcal{Y}$ an output space. Let $\mathfrak{P}_{\mathcal{X} \times \mathcal{Y}}$ be the set of all domains. We assume that we cannot observe domains directly. Instead, the training set consists of samples $\mathcal{S}$ taken from $N$ domains, $\mathcal{S} = \{S^{(d=i)}\}_{i=1}^N$. The $i$th domain $p^{(d=i)}(x, y)$ is represented by $n_i$ samples, $S^{(d=i)} = \{(x_k^{(d=i)}, y_k^{(d=i)})\}_{k=1}^{n_i}$. Here each of the $N$ distributions $p^{(d=1)}(x, y), \dots, p^{(d=i)}(x, y), \dots, p^{(d=N)}(x, y)$ is sampled from $\mathfrak{P}_{\mathcal{X} \times \mathcal{Y}}$. We further assume that $p^{(d=i)}(x, y) \neq p^{(d=j)}(x, y)$, therefore, the samples in $\mathcal{S}$ are non-i.i.d. During test time we are presented with samples $S^{(d=N+1)}$ from a previously unseen domain $p^{(d=N+1)}(x, y)$. We are interested in learning representations that generalize from $p^{(d=1)}(x, y), \dots, p^{(d=N)}(x, y)$ to this new domain. Training data are given as tuples $(d, x, y)$ in the case of supervised data or as $(d, x)$ in the case of unsupervised data.

### 2.2 DIVA: Domain Invariant VAE

Assuming a perfectly disentangled latent space (Higgins et al., 2018), we hypothesize that there exists a latent subspace that is invariant to changes in $d$, i.e., that is domain invariant. We propose a generative model with three independent sources of variation; $z_x$, which is object specific, $z_d$, which is domain specific and finally $z_y$, which is class specific. While $z_x$ keeps an independent Gaussian prior $p(z_x)$, $z_d$ and $z_y$ have conditional priors $p_{\theta_d}(z_d|d)$, $p_{\theta_y}(z_y|y)$ with learnable parameters $\theta_d, \theta_y$. This will encourage information about the domain $d$ and label $y$ to be encoded into $z_d$ and $z_y$ respectively. Furthermore, as $z_d$ and $z_y$ are marginally independent by construction, we argue that the model will learn representations $z_y$ that are invariant with respect to the domain $d$. All three of these latent variables are then used by a single decoder $p_\theta(x|z_d, z_x, z_d)$ for the reconstruction of $x$.

Since we are interested in using neural networks for $p_\theta(x|z_d, z_x, z_d)$, exact inference will be intractable. For this reason, we perform amortized variational inference with an inference network (Kingma & Welling, 2013), i.e., we employ a variational autoencoder (VAE) framework. We introduce three separate encoders $q_{\phi_d}(z_d|x)$, $q_{\phi_x}(z_x|x)$ and $q_{\phi_y}(z_y|x)$ that serve as variational posteriors over these latent variables. Notice that we do not share their parameters as we empirically found that sharing parameters leads to a decreased generalization performance. For the prior and variational posterior distributions over the latent variables $z_x, z_d, z_y$ we assume fully factorized Gaussians with parameters given as a function of their input. We coin the term domain invariant VAE (DIVA) for our overall model, which can be seen in Figure 1.

Given a specific dataset, all of the aforementioned parameters can then be optimized by maximizing the following variational lower bound:

$$
\begin{aligned}
\mathcal{L}_s(d,x,y) = \; & \mathbb{E}_{q_{\phi_d}(z_d|x)q_{\phi_x}(z_x|x),q_{\phi_y}(z_y|x)} \left[ \log p_\theta(x|z_d,z_x,z_y) \right] \\
& - \beta KL \left( q_{\phi_d}(z_d|x) || p_{\theta_d}(z_d|d) \right) - \beta KL \left( q_{\phi_x}(z_x|x) || p(z_x) \right) \\
& - \beta KL \left( q_{\phi_y}(z_y|x) || p_{\theta_y}(z_y|y) \right).
\end{aligned}
\tag{1}
$$

Notice that we have introduced a weigting term, $\beta$. This is motivated by the $\beta$-VAE (Higgins et al., 2017) and serves as a constraint that controls the capacity of the latent spaces of DIVA. Larger values of $\beta$ limit the capacity of each $z$ and in the ideal case each dimension of $z$ captures one of the conditionally independent factors in data. The $\beta$-VAE framework offers a trade-off between the information preservation, i.e., how well one can reconstruct $x$ from the $z$'s, and the capacity, i.e., how well the $z$'s compress information about $x$.

In order to further encourage separation of $z_d$ and $z_y$ into domain and class specific information respectively, we add two auxiliary objectives. During training $z_d$ is used to predict the domain $d$ and $z_y$ is used to predict the class $y$ for a given input $x$

$$
\mathcal{F}_{\text{DIVA}}(d,x,y) := \mathcal{L}_s(d,x,y) + \mathbb{E}_{q_{\phi_d}(z_d|x)q_{\phi_y}(z_y|x)} \left[ \alpha_d \log q_{\omega_d}(d|z_d) + \alpha_y \log q_{\omega_y}(y|z_y) \right], \tag{2}
$$

where $\alpha_d$, $\alpha_y$ are weighting terms for each of these auxiliary objectives. Since our main goal is a domain invariant classifier, during test time we are left with the encoder $q_{\phi_y}(z_y|x)$ and the auxiliary classifier $q_{\omega_y}(y|z_y)$. For predicting the class $y$ of a new input $x$ we only use the mean of $z_y$.

### 2.3 Guided Disentanglement

Locatello et al. (2018) and Dai & Wipf (2019) claim that learning a disentangled representation, i.e., $q_\phi(z) = \prod_i q_\phi(z_i)$, in an unsupervised fashion is impossible for arbitrary generative models. Inductive biases, e.g., some form of supervision or constraints on the latent space, are necessary to find a specific set of solutions that matches the true generative model. Consequently, DIVA is using domain labels $d$ and class labels $y$ in addition to input data $x$ during training. Furthermore, we enforce the factorization of the marginal distribution of $z$ in the following form: $q_\phi(z) = q_{\phi_d}(z_d)q_{\phi_x}(z_x)q_{\phi_y}(z_y)$, which prevents the impossibility described at Locatello et al. (2018). We argue that the strong inductive biases in DIVA make it possible to learn disentangled representations that match the ground truth factors of interest, namely, the domain $d$ and class $y$.

### 2.4 Semi-supervised DIVA

In (Kingma et al., 2014) an extension to the VAE framework was introduced that allows to use labeled as well as unlabeled data during training. While Kingma et al. (2014) introduced a two step procedure, Louizos et al. (2015) presented a way of optimizing the decoder of the VAE and the auxiliary classifier jointly. We use the latter approach to learn from supervised data $\{(d_n,x_n,y_n)\}$ as well as from unsupervised data $\{(d_m,x_m)\}$. Analogically to (Louizos et al., 2015), we use $q_{\omega_y}(y|z_y)$ to marginalize out $y$:

$$
\begin{aligned}
\mathcal{L}_u(d,x) = \; & \mathbb{E}_{q_{\phi_d}(z_d|x)q_{\phi_x}(z_x|x)q_{\phi_y}(z_y|x)}[\log p_\theta(x|z_d,z_x,z_y)] \\
& - \beta KL(q_{\phi_d}(z_d|x)||p_{\theta_d}(z_d|d)) - \beta KL(q_{\phi_x}(z_x|x)||p(z_x)) \\
& + \beta \mathbb{E}_{q_{\phi_y}(z_y|x)q_{\omega_y}(y|z_y)}[\log p_{\theta_y}(z_y|y) - \log q_{\phi_y}(z_y|x)] \\
& + \mathbb{E}_{q_{\phi_y}(z_y|x)q_{\omega_y}(y|z_y)}[\log p(y) - \log q_{\omega_y}(y|z_y)],
\end{aligned}
\tag{3}
$$

where we use Monte Carlo sampling with the reparametrization trick (Kingma & Welling, 2013) for the continuous latents $z_d, z_x, z_y$ and explicitly marginalize over the discrete variable $y$. The final objective combines the supervised and unsupervised variational lower bound as well as the two auxiliary objectives. By assuming $N$ labeled and $M$ unlabeled data tuples we arrive at the following objective

$$
\mathcal{F}_{\text{SS-DIVA}} = \sum_{n=1}^{N} \mathcal{F}_{\text{DIVA}}(x_n,y_n,d_n) + \sum_{m=1}^{M} \mathcal{L}_u(x_m,d_m) + \alpha_d \mathbb{E}_{q_{\phi_d}(z_d|x_m)}[\log q_{\omega_d}(d_m|z_d)]. \tag{4}
$$

# 3 RELATED WORK

**Domain generalization** The majority of recently proposed deep learning methods for domain generalization falls into one of two categories: 1) Learning a single domain invariant representation, e.g., using adversarial methods (Carlucci et al., 2018; Ghifary et al., 2015; Li et al., 2018; 2017; Motiian et al., 2017; Shankar et al., 2018; Wang et al., 2019). 2) Ensembling models, each trained on an individual domain from the training set (Ding & Fu, 2018; Mancini et al., 2018).

**Multi-task learning** Zamir et al. (2018) show that sharing parameters among multiple tasks can lead to better performance and can decrease the amount of labeled examples necessary for each task. Even though we make use of multiple tasks, our goal is the opposite. The three encoders of DIVA are forced to learn complementary features. For a general introduction to multi-task learning we refer to Ruder (2017).

**Fairness** The goal of fair classification is to learn good representation that at the same time cannot be used to associate a data sample to a certain group (Zemel et al., 2013). The only difference to domain generalization is the intention behind that goal, e.g., to protect groups of individuals *vs.* being robust to technical variations. Consequently, DIVA is closely related to the fair VAE (Louizos et al., 2015). The major difference lies in the fashion we replace the hierarchical latent space of the fair VAE with a partitioned latent space.

**Latent subspaces** A series of very recent papers demonstrate the benefits of having a partitioned latent space in a VAE. Klys et al. (2018) show that two latent supaces can be better disentangled using binary labels and adversarial training. In (Bouchacourt et al., 2018) two latent subspaces exist as well. Here one is used for individual samples and one is shared among samples from the same group. Last, Hsu & Glass (2018) show that latent subspaces can be used to capture information from different modalities (e.g., image and audio).

# 4 EXPERIMENTS

## 4.1 ROTATED MNIST

The construction of the rotated MNIST dataset follows (Ghifary et al., 2015). We sample 100 images from each of the 10 classes from the original MNIST training dataset (in contrast to (Ghifary et al., 2015) we do not resize the images). This set of images is denoted $\mathcal{M}_{0°}$. To create five additional domains the images in $\mathcal{M}_{0°}$ are rotated by 15, 30, 45, 60 and 75 degrees. In order to evaluate their domain generalization abilities models are trained on five domains and tested on the remaining 6th domain, e.g., train on $\mathcal{M}_{0°}$, $\mathcal{M}_{15°}$, $\mathcal{M}_{30°}$, $\mathcal{M}_{45°}$ and $\mathcal{M}_{60°}$, test on $\mathcal{M}_{75°}$. The evaluation metric is the classification accuracy on the test domain. All experiments are repeated 10 times with 10 different seeds, resulting in 10 different datasets. Detailed information about hyperparameters, architecture and training schedule can be found in the Appendix.

## 4.2 QUALITATIVE DISENTANGLEMENT

First of all, we visualize the three latent spaces $z_d$, $z_x$ and $z_y$, to see if DIVA is able to successfully disentangle them. In addition, we want to see if DIVA utilizes $z_x$ in a meaningful way, since it is not directly connected to any downstream task. For the following visualizations we restrict the size of each latent space $z_d$, $z_x$ and $z_y$ to 2 dimensions. Therefore, we can plot the latent supspaces directly without applying dimensionality reduction. DIVA is trained on 5000 images from five domains ($\mathcal{M}_{0°}$, $\mathcal{M}_{15°}$, $\mathcal{M}_{30°}$, $\mathcal{M}_{45°}$ and $\mathcal{M}_{60°}$).

Figure 2 shows 5000 embeddings $z_y$ encoded by $q_{\phi_y}(z_y|x)$. In Figure 2 (left) the colors indicate the 10 different classes of the MNIST dataset. We observe 10 well separated clusters, each corresponding to one of the 10 classes. In stark contrast, Figure 2 (right), where the colors indicate the five different training domains, shows no such clustering. It appears that $z_y$ is indeed capturing all necessary information for predicting the class $y$ while containing very little information about the domain $d$.

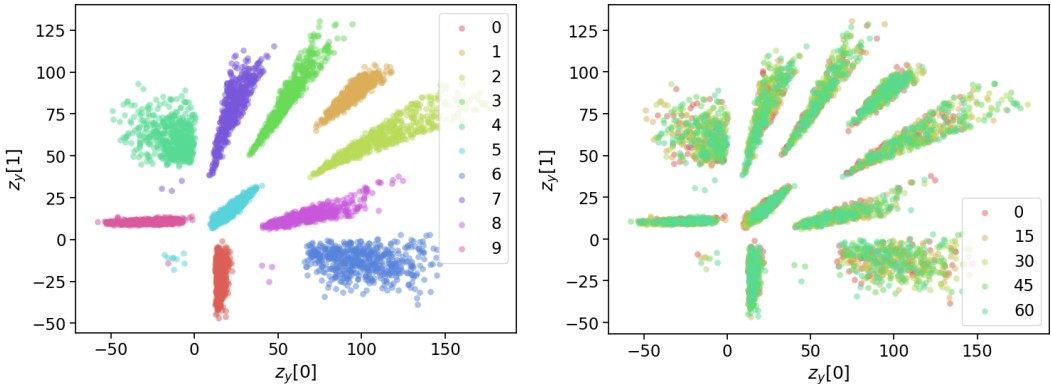

Figure 2: Left: 5000 two-dimensional embeddings $z_y$ encoded by $q_{\phi_y}(z_y|x)$. The color of each point indicates the associated class. Both dimensions of $z_y$ are used to encode the label. Right: Plot of the same embeddings as seen on the left. This time the color indicates the associated domain. No apparent clustering is visible.

In Figure 3 we visualize the two-dimensional latent space for $z_d$. Each of the 5000 training images is encoded by $q_{\phi_d}(z_d|x)$. Figure 3 (left) shows very little clustering according to the class label $y$. However, Figure 3 (right), where each color represents a different domain, shows strong clustering. Each cluster corresponds to one of the five domains.

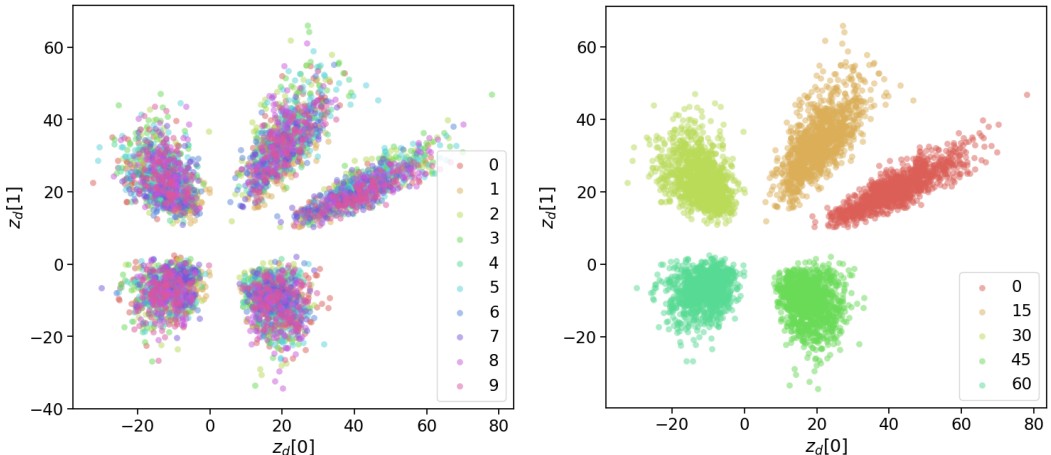

Figure 3: Left: 5000 two-dimensional embeddings $z_d$ encoded by $q_{\phi_d}(z_d|x)$. The color of each point indicates the associated class. No apparent clustering is visible. Right: Plot of the same embeddings as seen on the left. This time the color indicates the associated domain. The plot shows five distinct clusters, where each cluster corresponds to a single domain.

In contrast to $z_y$ and $z_d$, $z_x$ is only used for reconstructing $x$ and has an independent Gaussian prior. In Figure 4 (left) we can see that there is a certain amount of clustering. We find that narrow numbers with thin lines, e.g., '1' and '7', cluster in the bottom left half of the plot. Whereas round numbers with thicker lines, e.g., '0' and '6' appear to cluster in the top right half. We conclude that $z_x$ models the remaining variations that are not captured by $z_d$ and $z_y$. In Figure 4 (right) we do not notice any apparent clustering.

From these initial qualitative results we conclude that DIVA is disentangling the information contained in $x$ as intended, as $z_y$ is only containing information about $y$ and $z_d$ only information about $d$. In the case of the rotated MNIST dataset $z_x$ captures information about line thickness and digit width, two factors of variation that are not correlated with either the class $y$ or the domain $d$. We

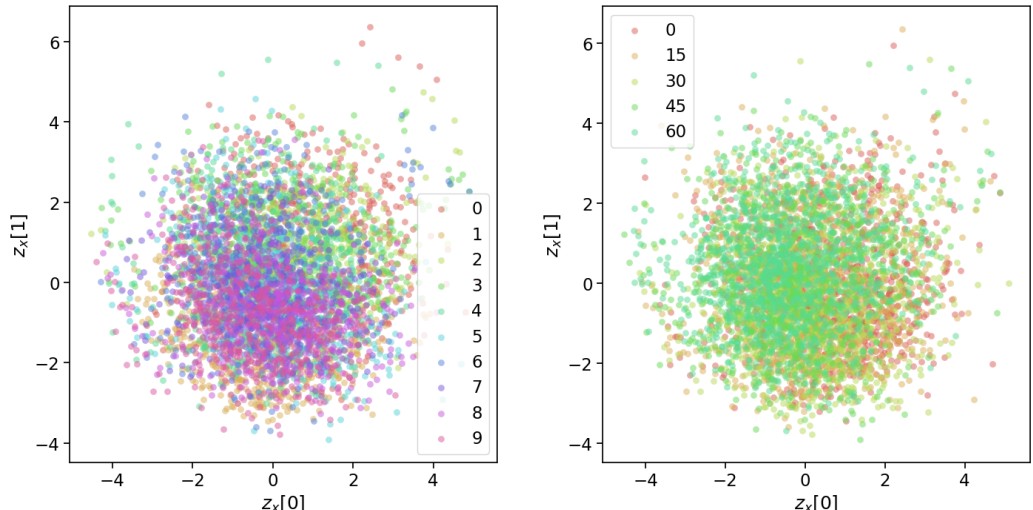

Figure 4: Left: 5000 two-dimensional embeddings $z_x$ encoded by $q_{\phi_x}(z_x|x)$. The color of each point indicates the associated class. We observe a high correlation between the line thickness of each MNIST digit and $z_x[0]$. Whereas, $z_x[1]$ is highly correlated with the width of each MNIST image. For that reason we observe a clustering of embeddings with class '1' at the lower half of the plot. Right: Plot of the same embeddings as seen on the left. This time the color indicates the associated domain. No apparent clustering is visible.

Table 1: We compare DIVA with other state-of-the-art domain generalization methods. Methods in the first half of the table (until the vertical line) use only labeled data. The second half of the table shows results of DIVA when trained semi-supervised (+ X times the amount of unlabeled data). We report the average and standard error of the classification accuracy.

| Test | DA | LG | HEX | ADV | DIVA | DIVA(+1) | DIVA(+3) | DIVA(+5) | DIVA(+9) |
|------|------|------|------|------|------|------|------|------|------|
| $\mathcal{M}_{0°}$ | 86.7 | 89.7 | 90.1 | 89.9 | **93.5 ± 0.1** | 93.8 ± 0.1 | 93.9 ± 0.2 | 93.2 ± 0.2 | 93.0 ± 0.1 |
| $\mathcal{M}_{15°}$ | 98.0 | 97.8 | 98.9 | 98.6 | **99.3 ± 0.1** | 99.4 ± 0.1 | 99.5 ± 0.1 | 99.5 ± 0.1 | 99.6 ± 0.1 |
| $\mathcal{M}_{30°}$ | 97.8 | 98.0 | 98.9 | 98.8 | **99.1 ± 0.1** | 99.3 ± 0.1 | 99.3 ± 0.1 | 99.3 ± 0.1 | 99.3 ± 0.1 |
| $\mathcal{M}_{45°}$ | 97.4 | 97.1 | 98.8 | 98.7 | **99.2 ± 0.1** | 99.0 ± 0.1 | 99.2 ± 0.1 | 99.3 ± 0.1 | 99.3 ± 0.1 |
| $\mathcal{M}_{60°}$ | 96.9 | 96.6 | 98.3 | 98.6 | **99.3 ± 0.1** | 99.4 ± 0.1 | 99.4 ± 0.1 | 99.4 ± 0.1 | 99.2 ± 0.1 |
| $\mathcal{M}_{75°}$ | 89.1 | 92.1 | 90.0 | 90.4 | **93.0 ± 0.1** | 93.8 ± 0.1 | 93.8 ± 0.1 | 93.5 ± 0.1 | 93.2 ± 0.1 |
| Avg | 94.3 | 95.3 | 95.8 | 95.2 | **97.2 ± 0.1** | 97.5 ± 0.1 | 97.5 ± 0.1 | 97.4 ± 0.1 | 97.3 ± 0.1 |

also can perform conditional reconstructions with DIVA. The results along with more details can be found in the Appendix.

### 4.3    COMPARISON TO OTHER METHODS

We compare DIVA against the well known domain adversarial neural networks (DA) (Ganin et al., 2015) as well as three recently proposed methods: LG (Shankar et al., 2018), HEX (Wang et al., 2019) and ADV (Wang et al., 2019).

For the first half of Table 1 (until the vertical line) we only use labeled data. The first column indicates the rotation angle of the test domain. We report test accuracy on $y$ for all methods. For DIVA we report the mean and standard error for 10 repetitions. DIVA achieves the highest accuracy across all test domains. In addition we achieve the highest average test accuracy among all proposed methods.

The second half of Table 1 showcases the ability of DIVA to use unlabeled data. For this experiment we add: The same amount (+1) of unlabeled data as well as three (+3), five (+5) and nine (+9) times the amount of unlabeled data to our training set. Here, we first add the unlabeled data to $\mathcal{M}_{0°}$ and create the data for the other domains as described in Section 4.1. In Table 1 we can clearly see a performance increase when unlabeled data is added to the training set. The effect seems to become

smaller when the amount of unlabeled data is much larger than the amount of labeled data as seen in the last two columns of Table 1.

## 4.4 ADDITIONAL UNLABELED DOMAINS

In Section 4.3 we show that the performance of DIVA increases when it is presented with additional unlabeled data for each domain. As a result each training domain consists of labeled and unlabeled examples. In this section we investigate a more challenging scenario: We add an additional domain to our training set that consists of only unlabeled examples. Coming back to our introductory example of medical imaging, here we would we add unlabeled data from a new patient or hospital to the training set. In contrast to the experiment in Section 4.3 where we would add unlabeled data from each known patient or hospital to the training set.

In the following, we are looking at two different experimental setups: 1) The additional domain is dissimilar to the test domain, e.g. $\mathcal{M}_{30°}$ and $\mathcal{M}_{75°}$. 2) The additional domain is similar to the test domain, e.g. $\mathcal{M}_{60°}$ and $\mathcal{M}_{75°}$. In both cases we show that DIVA improves when trained with unlabeled data from an unseen domain.

### 4.4.1 DISSIMILAR DOMAIN

For the first experiment we choose the domains $\mathcal{M}_{0°}$, $\mathcal{M}_{15°}$, $\mathcal{M}_{45°}$ and $\mathcal{M}_{60°}$ to be part of the labeled training set. In addition, unlabeled data from $\mathcal{M}_{30°}$ is used. The test domain is $\mathcal{M}_{75°}$. In Table 2 we can see that even in the case where the additional domain is dissimilar to the test domain DIVA is able to slightly improve.

Table 2: We compare DIVA trained with only labeled data to DIVA trained with additional unlabeled data from $\mathcal{M}_{30°}$. We report the average and standard error of the classification accuracy on $\mathcal{M}_{75°}$.

| Test | Only labeled data | Additional unlabeled |
|------|------|------|
| $\mathcal{M}_{75°}$ | $93.1 \pm 0.2$ | $93.3 \pm 0.1$ |

### 4.4.2 SIMILAR DOMAIN

For the second experiment we choose the domains $\mathcal{M}_{0°}$, $\mathcal{M}_{15°}$, $\mathcal{M}_{30°}$ and $\mathcal{M}_{45°}$ to be part of the labeled training set. In addition, unlabeled data from $\mathcal{M}_{60°}$ is used. The test domain is $\mathcal{M}_{75°}$. When comparing the results in Table 3 to the results in Table 1 and 2 we notice a drop in accuracy of about 20% for DIVA trained with only labeled data. However, when trained with unlabeled data from $\mathcal{M}_{60°}$ the drop in accuracy is only about 13%.

Table 3: We compare DIVA trained with only labeled data to DIVA trained with additional unlabeled data from $\mathcal{M}_{60°}$. We report the average and standard error of the classification accuracy on $\mathcal{M}_{75°}$.

| Test | Only labeled data | Additional unlabeled |
|------|------|------|
| $\mathcal{M}_{75°}$ | $73.8 \pm 0.3$ | $80.64 \pm 0.4$ |

## 5 CONCLUSION

We have proposed DIVA as a generative model with three independent sources of variation. We demonstrate through quantitative and qualitative experiments on rotated MNIST that our model successfully learns representations $z_y$ that are invariant with respect to the domain $d$. In future work, we want to evaluate DIVA on a more complex dataset.

ACKNOWLEDGMENTS

The authors thank Patrick Forré, Rianne van den Berg, Marco Federici, Daniel Worrall and Bas Veeling for helpful discussions and comments.

Maximilian Ilse was funded by the Nederlandse Organisatie voor Wetenschappelijk Onderzoek (Grant DLMedIa: Deep Learning for Medical Image Analysis).

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

## APPENDIX

### TRAINING PROCEDURE

All DIVA models are trained for 500 epochs. The training is terminated if the classification loss for $y$ has not improved for 100 epochs. As proposed in Burgess et al. (2018), we linearly increase $\beta$ from 0.0 to 1.0 during the first 100 epochs of training. We set $\alpha_d = 2000$. As seen in (Maale et al., 2019), we adjust $\alpha_y$ according to the ratio of labeled (N) and unlabeled data (M),

$$\alpha_y = \gamma \frac{M + N}{N}, \tag{5}$$

where we set $\gamma = 3500$. Last, $z_d$, $z_x$ and $z_y$ each have 64 latent dimensions. All hyperparameters were determined by training DIVA on $\mathcal{M}_{0°}$, $\mathcal{M}_{15°}$, $\mathcal{M}_{30°}$, $\mathcal{M}_{45°}$ and testing on $\mathcal{M}_{60°}$. All models were trained using ADAM (Kingma & Ba, 2014) (with default settings), a pixel-wise cross entropy loss and a batch size of 100.

### ARCHITECTURE

To enable a fair experiment, the encoder $q_{\phi_y}(z_y|x)$ and auxiliary classifier $q_{\omega_y}(y|z_y)$ form a DNN with the same number of layers and weights as described in Wang et al. (2019).

Table 4: Architecture for $p_\theta(x|z_d, z_x, z_y)$. The parameter for Linear is output features. The parameters for ConvTranspose2d are output channels and kernel size. The parameter for Upsample is the upsampling factor. The parameters for Conv2d are output channels and kernel size.

| block | details |
|-------|---------|
| 1 | Linear(1024), BatchNorm1d, ReLU |
| 2 | Upsample(2) |
| 3 | ConvTranspose2d(128, 5), BatchNorm2d, ReLU |
| 4 | Upsample(2) |
| 5 | ConvTranspose2d(256, 5), BatchNorm2d, ReLU |
| 6 | Conv2d(256, 1) |

Table 5: Architecture for $p_{\theta_d}(z_d|d)$ and $p_{\theta_y}(z_y|y)$. Each network has two heads one for the mean and one for the scale. The parameter for Linear is output features.

| block | details |
|-------|---------|
| 1 | Linear(64), BatchNorm1d, ReLU |
| 2.1 | Linear(64) |
| 2.2 | Linear(64), Softplus |

Table 6: Architecture for $q_{\phi_d}(z_d|x)$, $q_{\phi_x}(z_x|x)$ and $q_{\phi_y}(z_y|x)$. Each network has two heads one for the mean one and for the scale. The parameters for Conv2d are output channels and kernel size. The parameters for MaxPool2d are kernel size and stride. The parameter for Linear is output features.

| block | details |
|-------|---------|
| 1 | Conv2d(32, 5), BatchNorm2d, ReLU |
| 2 | MaxPool2d(2, 2) |
| 3 | Conv2d(64, 5), BatchNorm2d, ReLU |
| 4 | MaxPool2d(2, 2) |
| 5.1 | Linear(64) |
| 5.2 | Linear(64), Softplus |

Table 7: Architecture for $q_{\omega_d}(d|z_d)$ and $q_{\omega_y}(y|z_y)$. The parameter for Linear is output features.

| block | details |
|---|---|
| 1 | ReLU, Linear(5 for $q_{\omega_d}(d|z_d)$/10 for $q_{\omega_y}(y|z_y)$), Softmax |

## CONDITIONAL GENERATION

Yet another way to gain insight into the disentanglement abilities of DIVA is conditional generation. We first train DIVA with $\beta = 10$ using $\mathcal{M}_{0^\circ}$, $\mathcal{M}_{15^\circ}$, $\mathcal{M}_{30^\circ}$, $\mathcal{M}_{45^\circ}$ and $\mathcal{M}_{60^\circ}$ as training domains. After training we perform two experiments. In the first one we are fixing the class and varying the domain. In the second experiment we are fixing the domain and varying the class.

**Change of class** The first row of Figure 5 (left) shows the input images $x$ for DIVA. First, we generate embeddings $z_d$, $z_x$ and $z_y$ for each $x$ using $q_{\phi_d}(z_d|x)$, $q_{\phi_x}(z_x|x)$ and $q_{\phi_y}(z_y|x)$. Second, we replace $z_y$ with a sample $z'_y$ from the conditional prior $p_{\theta_y}(z_y|y)$. Last, we generate new images from $z_d$, $z_x$ and $z'_y$ using the trained encoder $p_\theta(x|z_d, z_x, z_y)$. In Figure 5 (left) rows 2 to 11 correspond to the classes '0' to '9'. We observe that the rotation angle (encoded in $z_d$) and the line thickness (encoded in $z_x$) are well preserved, while the class of the image is changing as intended.

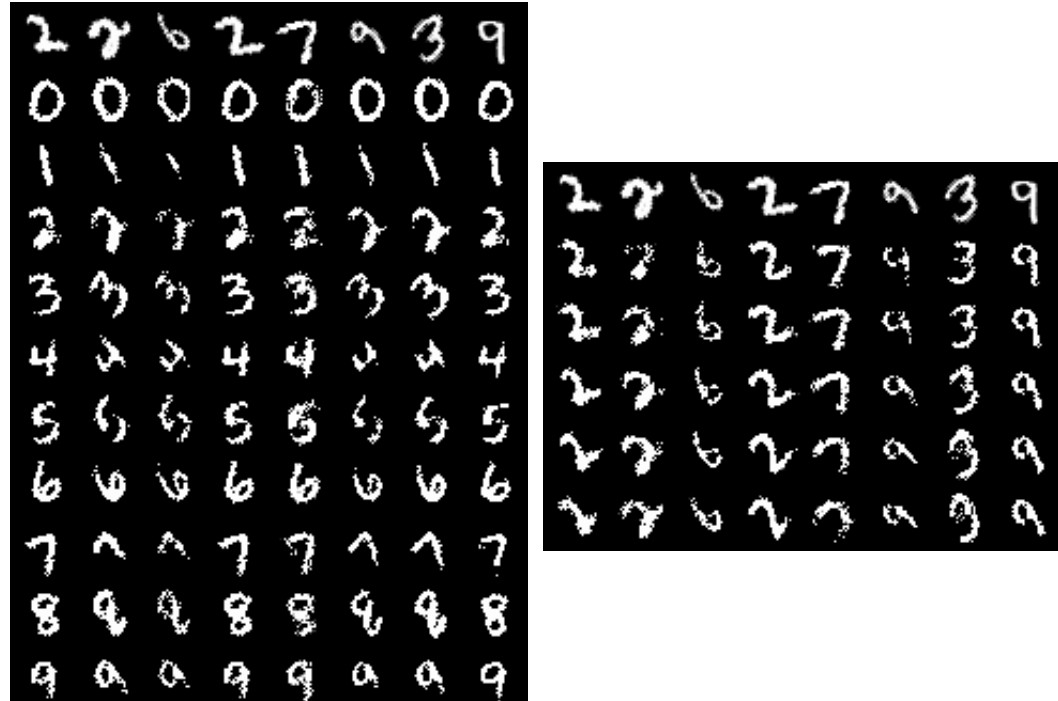

Figure 5: Reconstructions. Left: First row is input, row 2 to 11 correspond to labels '0' to '9'. Right: First row is input, row 2 to 6 correspond to domains 0, 15, 30, 45, 60.

**Change of domain** We repeat the experiment from above but this time we keep $z_x$ and $z_y$ fixed while changing the domain. After generating embeddings $z_d$, $z_x$ and $z_y$ for each $x$ in the first row of Figure 5 (right), we replace $z_d$ with a sample $z'_d$ from the conditional prior $p_{\theta_d}(z_d|d)$. Finally, we generate new images from $z'_d$, $z_x$ and $z_y$ using the trained encoder $p_\theta(x|z_d, z_x, z_y)$. In Figure 5 (right) rows 2 to 6 correspond to the domains $\mathcal{M}_{0^\circ}$ to $\mathcal{M}_{60^\circ}$. Again, DIVA shows the desired behaviour: While the rotation angle is changing the class and style of the original image is maintained.

QUALITATIVE DISENTANGLEMENT: TEST DOMAIN

In this section, we visualize the $z_d$ and $z_y$ for data points $x$ from the test domain $\mathcal{M}_{75°}$ for the model trained in Section 4.2. Figure 6 shows 1000 embeddings $z_y$ encoded by $q_{\phi_y}(z_y|x)$. Figure 7 shows 1000 embeddings $z_d$ encoded by $q_{\phi_d}(z_d|x)$.

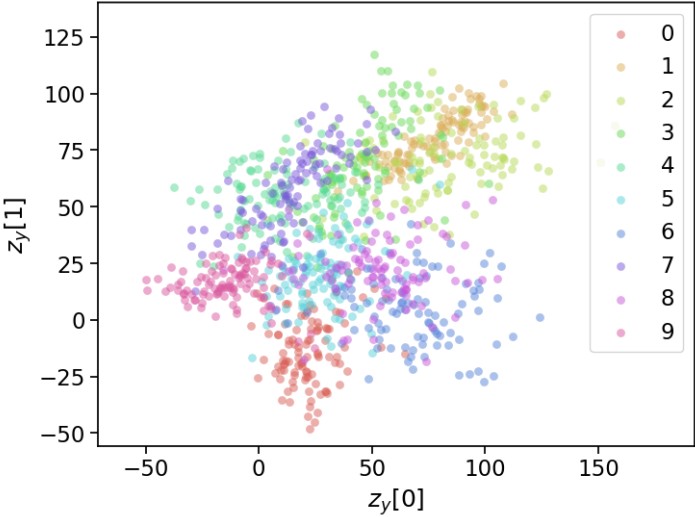

Figure 6: 1000 two-dimensional embeddings $z_y$ encoded by $q_{\phi_y}(z_y|x)$ for $x$ from the test domain $\mathcal{M}_{75°}$. The color of each point indicates the associated class.

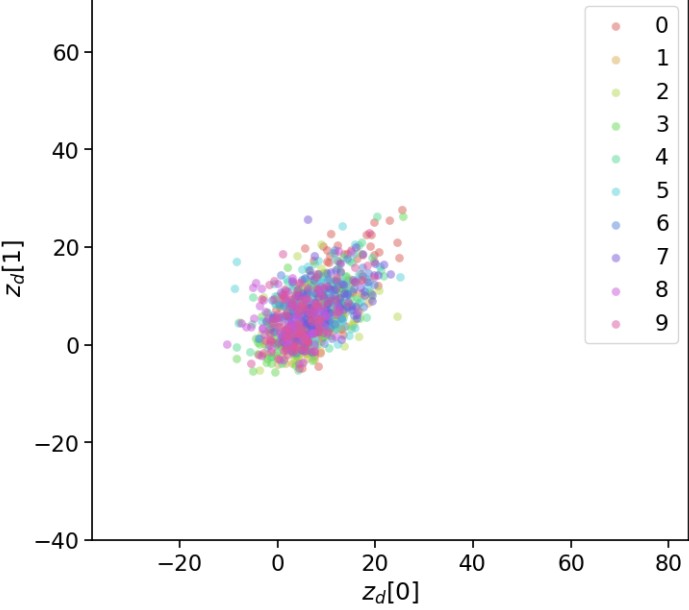

Figure 7: 1000 two-dimensional embeddings $z_d$ encoded by $q_{\phi_d}(z_d|x)$ for $x$ from the test domain $\mathcal{M}_{75°}$. The color of each point indicates the associated class.

