# OpenReview forum: "DIVA: Domain Invariant Variational Autoencoder"
_ICLR.cc/2019/Workshop/DeepGenStruct — DeepGenStruct 2019_

### Official Review · AnonReviewer1 · 2019-04-12
**Intuitive model design, but the experiments are kind of boring**

**Rating:** 3
**Confidence:** 2

**Review:**

The main contribution of this paper is the proposed Domain Invariant VAE (DIVA), which decomposes the latent space of a VAE into 3 parts: one for the class, one for the domain, and one for the rest. Additional auxiliary classifiers are introduced to encourage the separation of domain and class specific latent codes. This framework has also been extended to semi-supervised learning.

Pros: The studied problem is interesting, and the model itself is also intuitive. The overall framework looks elegant.

Cons:
(i) There is nothing special that surprises me in the model. It follows standard VAE design and standard extension to semi-supervised learning for VAE. It naturally extends the original VAE to incorporate domain-specific latent code inside. So I feel the whole model design is kind of boring.

(ii) Only experiments on MNIST are considered, so the experiments are relatively less interesting. As also noted by the authors, more interesting experiments on more challenging datasets are desired.

---

### Official Review · AnonReviewer2 · 2019-04-16
**need more experiments**

**Rating:** 3
**Confidence:** 2

**Review:**

The paper considers the problem of domain generalization: how to learn representations given data from a set of domains that generalize to data from a previously unseen domain.

The paper needs to better define "domain invariant". The rotated MNIST dataset is used for evaluation. But I do not think rotated images are from different domains. The paper needs more convincible experiments to prove the effectiveness of the proposed methods.

---

### Decision · Program_Chairs · 2019-04-19
**Acceptance Decision**

**Decision:**

Accept

**Comment:**

Accepted